# Optical properties of Central Asian aerosol relevant for spaceborne lidar applications and aerosol typing at 355 and 532 nm

Julian Hofer[1], Albert Ansmann[1], Dietrich Althausen[1], Ronny Engelmann[1], Holger Baars[1], Khanneh Wadinga Fomba[1], Ulla Wandinger[1], Sabur F. Abdullaev[2], and Abduvosit N. Makhmudov[2]

[1]Leibniz Institute for Tropospheric Research, Leipzig, Germany
[2]Physical Technical Institute of the Academy of Sciences of Tajikistan, Dushanbe, Tajikistan

**Correspondence:** hofer@tropos.de

**Abstract.**

For the first time, a dense data set of particle extinction-to-backscatter ratios (lidar ratios), linear depolarization ratios, and backscatter- and extinction-related Ångström exponents for a Central Asian site are presented. The observations were performed with a continuously running multiwavelength polarization Raman lidar at Dushanbe, Tajikistan, during an 18-month campaign (March 2015 to August 2016). The presented seasonally resolved observations fill an important gap in the data base of aerosol optical properties used in aerosol typing efforts with spaceborne lidars and ground-based lidar networks. Lidar ratios and depolarization ratios are also basic input parameters in spaceborne lidar data analyses and in efforts to harmonize long-term observations with different space lidar systems operated either at 355 or 532 nm. As a general result, the found optical properties reflect the large range of occurring aerosol mixtures consisting of long-range-transported dust (from the Middle East and the Sahara), regional desert, soil, and salt dust, and anthropogenic pollution. The full range from highly polluted to pure dust situations could be observed. Typical dust depolarization ratios of 0.23–0.29 (355 nm) and 0.30–0.35 (532 nm) were observed. In contrast, comparably low lidar ratios were found. Dust lidar ratios at 532 nm accumulated around 35–40 sr and were even lower for regional background dust conditions (20–30 sr). Detailed correlation studies (e.g., lidar ratio vs depolarization ratios and Ångström exponent vs lidar ratio and vs depolarization ratio) are presented to illuminate the complex relationships between the observed optical properties and to identify the contributions of anthropogenic haze, dust, and background aerosol to the overall aerosol mixtures found within the 18-month campaign. The observation of 532 nm lidar ratios ($<$25 sr) and depolarization ratios (around 15-20%) in layers with very low particle extinction coefficient ($<$30 sr) suggest that direct emission and emission of re-suspended salt dust (initially originated from numerous desiccating lakes and the Aralkum desert) have a sensitive impact on the aerosol background optical properties over Dushanbe.

## 1 Introduction

Central Asia is a hot spot region of severe environmental problems and climate-change effects. Shrinking glaciers and the desiccating Aral Sea are clear and unambiguous signs for a major impact of human activities on the environmental conditions in this area of the world (Kazakhstan, Turkmenistan, Uzbekistan, Kyrgyzstan, Tajikistan). Although a key region of strong

environmental changes, observations of aerosol conditions, one of the main indicators for the pollution state, are rare in Central Asia. Trustworthy data sets describing the annual cycle of environmental conditions as a function of height were absent until recently (see discussion in Hofer et al. (2020)).

Motivated by these observational gaps, we deployed a multiwavelength polarization Raman aerosol lidar at Dushanbe (38.5°N, 68.8°E, 864 m height above sea level, a.s.l.), Tajikistan, in the framework of the CADEX (Central Asian Dust Experiment) project (Althausen et al., 2019). The lidar (Polly: POrtabLe Lidar sYstem) (Engelmann et al., 2016; Baars et al., 2016) was continuously operated over a 18 months period from March 2015 to August 2016. First results were presented by Hofer et al. (2017). Profiles of basic aerosol optical properties and dust mass concentration in combination with vertically resolved dust source identification for representative aerosol scenarios were discussed based on case studies. Central Asia is in the middle of a large dust belt reaching from the western Sahara to the Gobbi desert in China. Long-range transport of mineral dust from the deserts in Middle East and from the Sahara, regional desert dust, local and regional anthropogenic haze and fire smoke, and background aerosol (mainly soil dust and salt dust from over 400 desiccating lakes in Central Asia and the Aralkum desert) lead to a complex aerosol mixture and complex vertical layering of aerosols in the planetary boundary layer and free troposphere, sometimes even up to heights close to the tropopause. A summary of the final results of this campaign in terms of aerosol height distributions, aerosol and dust optical thicknesses, height profiles of climate-relevant particle extinction coefficient at 532 nm, mass concentrations and mass fractions of dust and non-dust aerosol components, as well as of aerosol parameters influencing cloud evolution and precipitation formation such as cloud condensation nucleus and ice-nucleating particle concentrations was presented by Hofer et al. (2020). More than 300 individual (day by day) nighttime observations were analyzed and cover well the annual cycle of dust and aerosol pollution layering conditions.

In this study, we focus on the main findings regarding those particle optical properties that are used in aerosol-typing efforts with lidar networks and spaceborne lidars. Observations of particle extinction-to-backscatter ratios (lidar ratios), linear depolarization ratios at 355 and 532 nm wavelength, and backscatter-related and extinction-related Ångström exponents (describing the wavelength dependence of backscatter and extinction coefficients from 355 to 1064 nm), as presented here for Central Asian aerosol mixtures, are basic optical parameters in these aerosol-typing efforts and allow us to draw conclusions on the chemical composition (aerosol type), absorption and scattering properties, and size and shape characteristics of the particles (see, e.g., Müller et al., 2007; Burton et al., 2012, 2015; Groß et al., 2013; Nicolae et al., 2018; Papagiannopoulos et al., 2016, 2018; Voudouri et al., 2019). This article contributes to the steadily growing worldwide aerosol-typing data base.

Lidar ratios and depolarization ratios measured with ground-based lidar systems around the globe are also required to guarantee a quality-assured data analysis of spaceborne lidar measurements. Global aerosol monitoring is currently performed with NASA's spaceborne lidar CALIOP (Cloud Aerosol Lidar with Orthogonal Polarization) (Winker et al., 2009, 2010; Omar et al., 2009) of the CALIPSO (Cloud-Aerosol Lidar and Infrared Pathfinder Satellite Observations) mission and with ALADIN (Atmospheric LAser Doppler INstrument) of the Aeolus mission of the European Space Agency (ESA) (Stoffelen et al., 2005; Ansmann et al., 2007; Flamant et al., 2008; Lux et al., 2020; Witschas et al., 2020). In future, ATLID (Atmospheric Lidar) of the EarthCARE (Earth Clouds, Aerosol and Radiation Explorer) mission (Illingworth et al., 2015; Wandinger et al., 2016) will contribute to a 3-D global characterization of aerosols. Lidar ratios and depolarization ratios are input parameters in space lidar

data analyses operated at 532 and 1064 nm (CALIOP) and 355 nm (ALADIN, ATLID). Spectrally resolved lidar ratios and depolarization ratios, as presented here for Central Asian aerosol, are especially valuable with respect to the homogenization and harmonization of the overall 20-year CALIOP-ALADIN-ATLID aerosol climatology.

The article is structured as follows: In Sect. 2, we provide a brief overview of the CADEX campaign, the multiwavelength lidar, and observational products and uncertainties. Sections 3 and 4 contain the main findings and discussion regarding occurring aerosol types and origin of the aerosol particles forming the found aerosol mixtures. In Sect. 5, we discuss the potential impact of salt dust on the background aerosol optical properties. Concluding remarks are given in Sect. 6.

## 2   CADEX lidar data analysis

During the 18-month CADEX campaign from 17 March 2015 to 31 August 2016, a Polly-type multiwavelength polarization/Raman lidar (Engelmann et al., 2016; Hofer et al., 2017, 2020) was operated in Dushanbe, Tajikistan. The Dushanbe lidar station is part of PollyNET, a network of permanent or campaign-based Polly lidar stations (Baars et al., 2016) and is the first outpost of the European Aerosol Research Lidar Network (EARLINET) (Pappalardo et al., 2014). The polarization Raman lidar permits us to measure height profiles of the particle backscatter coefficient $\beta$ at the laser wavelengths of 355, 532 and 1064 nm, particle extinction coefficient $\alpha$ at 355 and 532 nm by means of 387 and 607 nm nitrogen Raman signal profiling, the particle linear depolarization ratio $\delta$ at 355 and 532 nm from the cross-polarized and total (co- and cross-polarized) lidar return signals at 355 and 532 nm, and of the water-vapor-to-dry-air mixing ratio by using the Raman lidar return signals at 407 nm (water vapor channel) and of the 387 nm nitrogen Raman channel (e.g., Baars et al., 2016; Engelmann et al., 2016; Hofer et al., 2017; Dai et al., 2018). The Raman lidar method (Ansmann et al., 1992) was used to compute the particle backscatter coefficients not only at 355 and 532 nm, but also at 1064 nm. Here, the data analysis is based on the height profile of the ratio of the 1064 nm to 607 nm nitrogen Raman signal (Mattis et al., 2004). Particle extinction and transmission effects at 532, 607, and 1064 nm are corrected by means of the available extinction coefficient profiles at 355 and 532 nm and the respective wavelength dependence. The advantage of the Raman lidar method is that lidar ratios are not needed as input as is the case in the elastic-backscatter lidar retrieval (Fernald, 1984; Klett, 1985).

The particle linear depolarization ratio is defined as

$$\delta = \frac{\beta_\perp}{\beta_\parallel} \, . \tag{1}$$

A polarization lidar transmits linearly polarized laser pulses and detects the cross- and co-polarized signal components, or in the case of Polly the cross-polarized and total backscatter signals, from which the volume (particle + Rayleigh) linear depolarization ratio and the particle depolarization ratio as defined in Eq. (1) can be determined. The specific depolarization ratio retrieval in the case of the Polly instrument is described in detail by Engelmann et al. (2016). The depolarization ratio is an important parameter and allows us to unambiguously identify non-spherical dust particles and to quantify the dust mass fraction defined as the ratio of dust-to-total-particle mass concentration (Hofer et al., 2020). The particle linear depolarization

ratio is around 0.25 at 355 nm and 0.30–0.35 at 532 nm for coarse-mode-dominated size distributions of mineral dust particles. For small, spherical particles (fine-mode particles) the depolarization ratio is usually <0.05 at both wavelengths.

The multiwavelength Raman lidar option permits the independent determination of the particle backscatter and extinction coefficients (Ansmann et al., 1992) at 355 and 532 nm and thus, the determination of the particle lidar ratio defined as

$$S = \frac{\alpha}{\beta}. \tag{2}$$

The lidar ratio depends on the size, shape and absorption and scattering properties of the particles, and can be as low as 15 sr for marine sea salt particles and >100 sr for strongly absorbing black-carbon-containing smoke particles. The lidar ratio retrieval has the highest priority and therefore we measure the total backscatter signal rather than the cross-and co-polarized backscatter signal components from which the total backscatter signal must then be constructed. By measuring the total lidar return signal we keep the uncertainties in the lidar ratio calculations as small as possible.

The measurement of the particle backscatter and extinction coefficients at several wavelengths $\lambda$ allows us to calculate the backscatter-related Ångström exponents $\mathring{a}_{\beta,355/532}$ and $\mathring{a}_{\beta,532/1064}$,

$$\mathring{a}_{\beta,\lambda_1/\lambda_2} = \frac{\ln(\beta_{\lambda_1}/\beta_{\lambda_2})}{\ln(\lambda_2/\lambda_1)}, \tag{3}$$

and the extinction-related Ångström exponent $\mathring{a}_{\alpha,355/532}$,

$$\mathring{a}_{\alpha,\lambda_1/\lambda_2} = \frac{\ln(\alpha_{\lambda_1}/\alpha_{\lambda_2})}{\ln(\lambda_2/\lambda_1)} \tag{4}$$

The Ångström exponent (Ångström, 1964) describes the spectral dependence of the optical properties and is strongly influenced by the size of the particles. The values are large (1.5–2) for fine-mode-dominated aerosols such as urban haze or biomass burning smoke consisting of particles with radius mainly <1 μm, and is low (0.5 to −1) for coarse-mode-dominated particle size distributions (desert dust, sea salt aerosol). The backscatter and extinction Ångström exponents are linked via (Ansmann et al., 2002),

$$\mathring{a}_S = \mathring{a}_\alpha - \mathring{a}_\beta, \tag{5}$$

with the Ångström exponent $\mathring{a}_S$ for the lidar ratio.

As outlined by Hofer et al. (2020), the polarization lidar allows us to separate the dust and non-dust-related backscatter, extinction and (by using conversion factors) mass concentration contributions (Mamouri and Ansmann, 2017). The technique was recently updated with focus on global desert dust sources (Ansmann et al., 2019). After separation, the dust mass fraction can be computed and used as an indicator for the aerosol mixing state (see Sect. 5).

## 3 Observations

During the 18-month CADEX campaign (535 days), the Polly lidar acquired data at 487 days for at least a 3 h time period. To achieve a representative coverage of aerosol conditions, profiles were calculated on a day by day basis for each night at

which the application of the Raman lidar methods was possible, i.e., when low clouds and fog were absent. The collected signal profiles were averaged over 60–180 minutes and smoothed with window lengths of typically 180–300 m (backscatter coefficients, depolarization ratios) and 750 m (extinction coefficients, lidar ratio). Raman lidar profiles of the particle extinction and backscatter coefficients and extinction-to-backscatter ratio (lidar ratio) at 355 and 532 nm could be obtained for 276 nights.

In further 52 night, we could make use of the Raman lidar method to determine the backscatter coefficient profile. However, the atmospheric conditions were not favorable to derive extinction and lidar ratio profiles as well.

As shown in Hofer et al. (2020), we analyzed the annual cycle of aerosol layering in the lower, middle, and upper troposphere over Dushanbe based on these 328 aerosol backscatter profiles. By visual inspection we found two main regimes: (a) the main aerosol layer that typically extends from the surface to about 3–6 km height and contributes to 500 nm aerosol optical thickness

(AOT) by usually more than 90%, and (b) frequently occurring thin dust layers between 5 and 10 km height that mainly contained aerosol from remote source regions such as the Arabian deserts and Saharan desert. In this article, we concentrate on the mean optical properties in the main aerosol layer.

### 3.1   Example of data analysis

Figure 1 shows an example measurement to illustrate the procedure how we determined the layer-mean values. An averaging

height range from 1.5–3 km height was chosen in this example and the layer mean optical properties were computed from the height profiles of the shown backscatter and extinction coefficients, depolarization ratios, lidar ratios, and Ångström exponents. The layer mean values were then used in the statistical analysis in Sect. 3.2 and in the correlation studies in Sect. 4.

Mineral dust dominated the optical properties on the 11 June 2015. Almost wavelength-independent backscatter and extinction coefficients in the short wavelength range (355–532 nm) were found as can be seen in Fig. 1a. The layer mean particle

depolarization ratios were higher than 0.20 (355 nm) and 0.30 (532 nm) and the mean lidar ratios were around 46 sr (355 nm) and 37 sr (532 nm). All this indicates dusty condition on 11 June 2015.

### 3.2   Statistical analysis

Figures 2 and 3 provide an overview of the depolarization and lidar ratios observed during the 18 months from March 2015 to August 2016. The data are shown for the different seasons. Two spring and summer seasons and one autumn and winter

seasons are covered. A broad distribution of depolarization ratios is found during the summer half year (spring to autumn). The same holds for the lidar ratios. These broad distributions are indicative for the occurrence of very different aerosol conditions with complex aerosol mixtures of anthropogenic and natural aerosol types and components.

According to Rupakheti et al. (2019), the major air pollution sources in Central Asia include lowgrade coal consumption in the households, industrial, construction, and mining activities, transportation facilities consisting of mostly old and untimely-

maintained vehicles, and natural sources such as agricultural activities and emissions from cropland, wildfires, deserts and desiccating lakes and seas (e.g., Aral Sea). Tajikistan lies in the proximity of the deserts like Taklamakan, Aralkum, Kyzylkum, Karakum and other deserts in Iran and Afghanistan. Desertification and the drying of the Aral Sea belong to the major environmental issues in Central Asia. Li and Sokolik (2017) estimated that the Aralkum desert contributes to 12% to the Central Asian

dust emissions. High aerosol loading during the spring and summer season is also favored by the low number of precipitation events. In addition to all the local and regional aerosol sources, long-range transport of dust and pollution contributes to the complexity of the observed aerosol mixtures (Hofer et al., 2020).

As can be seen in Fig. 2, the depolarization ratio ranged from values for pure urban haze conditions (0.0 to 0.03) to values typical for heavy dust outbreak conditions (up to 0.3 at 355 nm and up to 0.35 at 532 nm). Lidar ratios from 25–65 sr were found, accumulating around 30–45 sr (355 nm, mean of 42 sr) and 25–35 sr (532 nm, mean of 33 sr, see Fig. 3). In the summer half year, long-range transport of dust from southwesterly directions prevail (Hofer et al., 2020). Local and regional dust contribute in addition. The influence of anthropogenic pollution (industry, traffic) is low. Only during the winter months (heating season), urban haze dominates in Dushanbe. In winter, the impact of long-range transport of dust is strongly reduced because of low convective activity and thus, obviously also emissions over the desert areas.

The comparably low lidar ratios reflect the frequent occurrence of dust from Middle East and Central Asian desert regions. Lidar ratios for these dust particles are much lower (Mamouri et al., 2013; Nisantzi et al., 2015; Hofer et al., 2017; Filioglou et al., 2020) than the ones for central and western Saharan regions which are typically in the range of 40–65 sr (Mattis et al., 2002; Mona et al., 2006; Guerrero-Rascado et al., 2009; Giannakaki et al., 2010; Groß et al., 2011; Tesche et al., 2011; Kanitz et al., 2013; Preißler et al., 2013; Guerrero-Rascado et al., 2009; Veselovskii et al., 2016; Rittmeister et al., 2017; Bohlmann et al., 2018).

Figure 4 shows histograms of extinction-related Ångström exponents for the different seasons of the year. Again, a broad distribution of values from around 0 (pure dust) to 2.4 (fresh urban haze and smoke) are found. However, the seasonal mean values for the Ångström exponent (355–532 nm spectral range) are all at around 1.1±0.5.

Figure 5 summarizes the statistical analysis and provides insight into the annual cycle of the different optical properties found in the lower troposphere (see layer height information in Fig. 5e). Seasonal mean values, standard deviation, and range of values for the derived lidar ratios, depolarization ratios, and Ångström exponents are given in Tables 2 to 4.

Table 2 contrasts dust and non-dust optical properties. The used criterion for pure dust is $\delta > 0.31$ at 532 nm, and $\delta < 0.05$ for (non-dust) anthropogenic pollution cases. The mean values of the dust lidar ratios and depolarization ratios in Table 2 are very close to the values recently presented by Filioglou et al. (2020) for Middle East desert dust, measured at the emirate of Sharjah during a one-year campaign.

Table 3 summarizes seasonal mean values, standard deviations, and range of values for the lidar ratios and depolarization ratios at both wavelengths. The seasonal mean lidar ratio at 355 nm does not vary much from season to season, in contrast to the seasonal mean lidar ratio at 532 nm. During the summer half year, the seasonal mean values are 31–33 sr at 532 nm and thus even lower than the pure dust-related value of 39 sr in Table 2. When anthropogenic particles dominate (in winter), the mean value was found to be 43 sr at 532 nm (and thus, close to the value at 355 nm). These haze-related values are again lower than the respective pure pollution values in Table 2, and thus, point also to the presumption of a background aerosol causing a very low lidar ratio at 532 nm. This aspect is further discussed at the end of Sect. 4. The contrast between summer and winter depolarization values in Table 3 reflects the change from dust-dominated to pollution-dominated aerosol conditions as well.

Table 4 summarizes the findings concerning the backscatter- and extinction-related Ångström exponents. In the case of strong dust outbreaks with a large coarse-dust fraction, the Ångström exponents for the 355–532 nm spectral range are partly negative. Similar results are reported for Saharan dust when the observations were conducted within or close to the dust source region (Kanitz et al., 2013; Veselovskii et al., 2016; Rittmeister et al., 2017).

The observations at Dushanbe fit well into the general picture of lidar-derived particle optical properties for northern hemispheric aerosol mixtures. Overviews of lidar ratios and depolarization ratio for a variety of different aerosol types are given by, e.g., Müller et al. (2007) and Nicolae et al. (2018). We checked the literature especially with focus on dust-related optical properties. The numerous lidar observations of depolarization ratios around the world indicate quite uniform light depolarization characteristics for mineral dust. An overview is given by Shin et al. (2018). Maximum values of particle linear polarization ratios are 0.23–0.30 at 355 nm and 0.3–0.35 at 532 nm (e.g., Sugimoto and Lee, 2006; Freudenthaler et al., 2009; Burton et al., 2015; Veselovskii et al., 2016; Hofer et al., 2017; Hu et al., 2019; Filioglou et al., 2020). Sun-photometer-based studies corroborate our impression that light depolarization by dust is not a function of the dust source and does not vary much around the globe (Shin et al., 2018). Since the depolarization ratio is strongly influenced by particle shape and size distribution characteristics, we may conclude that natural mineral dust particle ensembles show rather similar size and shape properties disregarding the region where the dust particles were released into the atmosphere. The same conclusions were drawn by Ansmann et al. (2019).

However, a certain variability in the depolarization ratio characteristics must be always kept in consideration because of changes in the dust size distribution during long-range transport related to a stronger removal of coarse and giant particles from the atmosphere than the deposition of fine-mode dust particles. It was shown that the fine-mode dust fraction (particles with radius <500 nm) contribute about 20% to the overall dust AOT at 355 and 532 nm (Mamouri and Ansmann, 2014, 2017). We concluded from our combined photometer-lidar observations that the same fine-mode impact holds for the backscatter coefficients and depolarization ratios. The fine-mode related dust depolarization ratio is roughly 0.20, 0.15, and 0.10 for 355, 532, and 1064 nm wavelength, respectively. The coarse-mode causes depolarization ratios of around 0.25–0.30 (355 nm), 0.35–0.40 (532 nm), and 0.25–0.30 (1064 nm), as lidar observations of long-range transport of dust together with the laboratory studies indicate (Haarig et al., 2017a; Mamouri and Ansmann, 2017). This has to be taken into account in the comparison of the findings discussed in Sect. 4, and explains small changes in the dust depolarization ratios downwind the dust sources as found by Haarig et al. (2017a) and Rittmeister et al. (2017).

In contrast to these small depolarization ratio changes, the extinction-to-backscatter ratio varies considerably from dust source to dust source as a result of changing dust mixtures and thus changing chemical compositions. Latest overviews of measured dust lidar ratios are given by Nicolae et al. (2018), Shin et al. (2018), and Kim et al. (2020). Sakai et al. (2002) and Liu et al. (2002) for Asian dust and Mattis et al. (2002) for Saharan dust were the first who demonstrated that the dust lidar ratio is not close to 20 sr (as modelled for spherical dust particles), but in the range of 40–50 sr (532 nm, Asian dust) and 50–70 sr (355 and 532 nm, Saharan dust) because of the irregular shape of the dust particles. Mamouri et al. (2013) then emphasized that Middle East dust can cause a quite low lidar ratio of 34–39 sr (532 nm) and that this contrast to Saharan dust is probably

related to the difference in mineralogical composition. The low lidar ratios were also observed by Nisantzi et al. (2015) and Hofer et al. (2017).

Veselovskii et al. (2020) relate variations of the ratio of Saharan dust lidar ratios at 355 and 532 nm wavelength to the dust source region mineralogy. The ratio of the lidar ratios can be an indicator of an increased imaginary part of the complex refractive index in the UV because the lidar ratios are sensitive to the imaginary part of the complex refractive index. This could enable to discriminate different dust types of similar large depolarization ratios of >0.3 based on the ratio of the lidar ratios.

Shin et al. (2018) discussed the chemical composition of Saharan, Middle East, Asian, and Australian dust and the link to the characteristic lidar ratios. They stated that Saharan dust contains a comparably high amount of clay minerals (e.g., kaolinite, illite, montmorillonite) which absorbs light at solar wavelengths. The imaginary part of the refractive index increases towards shorter wavelengths. That could explain the relatively high Saharan dust lidar ratios. Varying contents of iron oxide minerals (e.g., hematite) which also strongly absorb in the UV and at visible wavelengths and varying amounts of calcite and gypsum which show almost no absorption in the UV and at visible wavelengths modulate the observed lidar ratio variations. According to Shin et al. (2018), lower portions of clay in the mineral compositions in eastern Asian and Middle East dust lead to lower lidar ratios (35–45 sr) compared to Saharan dust lidar ratios. Quartz has strong absorption bands in the IR, while its absorption properties are negligible at UV and visible wavelengths. Dust from Australian deserts consists mostly of quartz, thus the lidar ratios are in the range of 30–35 sr, 532 nm (Shin et al., 2018). All this is in consistency with the global lidar ratio study recently presented by Kim et al. (2020).

## 4   Correlation studies

In this section, we illuminate the relationship between the different lidar-derived aerosol optical properties to provide further insight into the aerosol mixing states and characteristics. The discussion is complementary to the findings presented by Hofer et al. (2020) and can be regarded as a contribution to the aerosol typing and classification efforts.

Figure 6 shows the basic correlation between the particle extinction coefficients at 355 and 532 nm and the respective 355 and 532 nm lidar ratios and depolarization ratios. In addition, the correlations between the extinction-related Ångström exponent (for the 355–532 nm wavelength range) and the extinction coefficient at 355 nm (in blue) and at 532 nm (in green) is shown. The shifts between the blue and and green data sets are partly caused by the different extinction coefficients. On average, the 355 nm extinction coefficients are roughly a factor of 1.5 larger than the 532 nm extinction coefficients as a result of the impact of non-dust aerosol (mainly fine-mode particles) leading to a mean Ångström exponent around 1 (see Fig. 4).

As can be seen in Fig. 6, the lidar ratio and depolarization ratio increase with increasing dust impact, indicated by an increasing extinction coefficient, whereas the Ångström exponent decreases and shows a broad spectrum from 0.5 to 2.2 for 532 nm extinction values <100 $\mathrm{Mm}^{-1}$. There is a clear shift in the lidar ratio and depolarization ratio data sets (355 nm vs 532 nm). In full agreement with many other studies (e.g. Groß et al., 2011; Haarig et al., 2017a), the 355 nm depolarization ratios are lower by about 0.1 disregarding the mixing state, mainly as the result of the generally stronger impact of fine-mode

particles on the 355 nm optical properties. The lidar ratios are larger at 355 nm because of the stronger impact of fine-mode aerosol pollution on the 355 nm extinction coefficient and the usually higher absorption of radiation. The lidar ratios are on average about 10 sr larger at 355 nm than at 532 nm.

For low extinction coefficients, the optical properties of the background aerosol become visible. At low overall particle extinction levels, neither dust outbreak nor local pollution conditions dominate. With decreasing particle extinction coefficient the 532 nm lidar ratio decreases towards 20 sr. The lidar ratio for 355 nm is higher because still influenced by residual haze in the background aerosol mixtures. For low extinction values, the noise (uncertainty) in the determined Ångström exponents becomes dominant so that clear conclusions regarding background Ångström values cannot be drawn. However, a broad distribution from 0.5–2.0 is visible for low 532 nm extinction values ($<25$ Mm$^{-1}$). The depolarization ratios do not show a clear picture but suggest that the background aerosol still contains a fraction of non-spherical particles. The 532 nm lidar ratios and depolarization ratios for background extinction levels may thus point to an increasing impact of dry salt particles to the light extinction (Haarig et al., 2017b) as discussed at the end of this section.

In Fig. 7, the correlation between the extinction-related Ångström exponent and the 355 and 532 nm lidar ratios and depolarization ratios are shown. An increasing Ångström exponent indicates an increasing fraction of anthropogenic haze and biomass burning smoke in the aerosol mixture. A well-defined relationship is found for the depolarization ratios. With decreasing Ångström exponent (and increasing influence of dust), the depolarization ratios increase and show maximum values of 0.25–0.3 (355 nm) and around 0.35 (532 nm). The correlation is less clear for the lidar ratio. The 355 nm extinction-to-backscatter ratio is always around 40 sr at 355 nm disregarding whether pollution or dust dominates. Also at 532 nm, the correlation is weak.

The impact of the given aerosol mixture on the lidar ratio is further illuminated in Fig. 8. Here, the depolarization ratio is plotted vs the lidar ratio. The individual (color-coded) measurements are sorted by dust mass fraction as well. As introduced in Sect. 2, the dust mass fraction describes the relative contribution of dust to the overall particle mass concentration. For dust fractions around 1 and depolarization ratios $>0.2$ (355 nm) and $>0.3$ (532 nm), the respective dust lidar ratio accumulate in the range of 40–45 sr (355 nm) and 30–40 sr (532 nm). In contrast, for low depolarization ratios and comparably low dust fractions, a large spread in the lidar ratio values is found (from 25–70 sr at 355 nm, and 20–60 sr at 532 nm). Note that the particle extinction coefficient for a given particle mass concentration is about a factor of 5 higher in the case of anthropogenic particles compared to dust particles. Thus, the dust mass fractions of 0.5 correspond roughly to a dust fraction of 0.1 in terms of dust and pollution extinction coefficients. Note also, that for low 532 nm lidar ratios ($<30$ sr, frequently found during aerosol background condition), many depolarization values are considerably higher than 0.1 and indicate the presence of a considerable fraction of non-spherical particles.

The curved feature in Fig. 8b with higher lidar ratios for small and large 532 nm depolarization ratios is in principle also found for 355 nm, however the range of observed depolarization ratio is smaller so that the curved feature is compressed and a clear minimum of the 355 nm lidar ratio for moderate depolarization ratios around 0.1 is not visible. Furthermore, the optical properties at 355 nm are dominated by scattering and absorption by fine-mode aerosol (especially by anthropogenic haze) and

are less influenced by scattering by coarse-mode (desert or salt) dust particles than the ones for the 532 nm wavelength. Thus, the background aerosol effect shows up more pronounced for 532 nm.

Figure 9 provides insight into the spectral dependence of the particle extinction coefficient vs the spectral dependencies of the depolarization ratio and lidar ratio. Again, the individual observations are color-coded according to the different dust mass fraction levels. Such a correlation is helpful in the discussion of data harmonization efforts when one spaceborne lidar data set is collected at 355 nm (ALADIN, ATLID) and the other data set at 532 nm. As can be seen, there is a weak correlation between the Ångström exponent and the ratios of depolarization and lidar ratios. For low values of the Ångström exponent (and high values of the dust mass fraction), the ratios $\delta_{355}/\delta_{532}$ and $S_{355}/S_{532}$ are approaching the pure dust values of around 0.75 and 1–1.2, respectively, whereas for Ångström exponents indicating dominating anthropogenic haze conditions the depolarization and lidar ratios are very variable and accumulate at 0.4–0.6 and 1.2–1.7, respectively.

To complete the discussion on the correlations between the lidar-derived optical properties, Fig. 10 shows a correlation of dust extinction coefficient (in contrast to Fig. 6 now only for 532 nm) and the backscatter-related Ångström exponents for the two wavelength ranges from 355–532 nm and 532–1064 nm. Mean values, standard deviations, and range of values are given in Table 4. For large extinction coefficients and thus a strong dust impact, we frequently found negative 355–532 nm backscatter-related Ångström exponents. This is a unique feature of dust, probably related to strong dust outbreaks (and dust observations close to the dust source region) with a pronounced coarse-mode dust fraction and the presence of even some giant particles (with diameters >20 µm). This feature was also observed, e.g., by Bohlmann et al. (2018) and Rittmeister et al. (2017) and especially emphasized by Veselovskii et al. (2016). The range of observed values is larger (−0.5 to 1.8) for the 355–532 nm backscatter-related Ångström exponents compared to the 532–1064 nm backscatter-related Ångström exponent (0.1 to 1.4). This corresponds to the different sensitivities of the three wavelengths to fine-mode particles, and large, and very large (or giant) coarse-mode particles (Mamouri and Ansmann, 2017; Haarig et al., 2017a). The likewise small range of 532–1064 nm backscatter-related Ångström exponents indicate that in most cases, however, very large or giant dust particles were absent over Dushanbe, removed during the long-range transport of dust.

## 5  Background aerosol: the potential impact of salt dust

On 45 days (15% out of the 276 cases of daily nighttime aerosol Raman lidar measurements), we found aerosol layers in the lower troposphere that showed a rather low particle extinction coefficient at 532 nm ($<30$ Mm$^{-1}$) together with an enhanced particle depolarization ratio of 10-20% and a lidar ratio of 25 sr and less. An example is shown in Fig. 11. Local and regional pollution influenced the optical properties in the lowest 1.5-1.6 km height. However, above 1.6 km a relatively clean layer with background aerosol and particle extinction coefficients around 20 Mm$^{-1}$ was observed.

The corresponding backward trajectories for the central height range of the entire aerosol layer (up to 2.5 km height) are shown in Fig. 12. The trajectories indicate an airflow from western to northern directions. They partly crossed the Caspian Sea, the saline lake Garabogazköl (forming a lagoon of the Caspian Sea), and the Aral Sea (now Aralkum desert). We hypothesize that the background aerosol, detected between 1.6 and 2.5 km height, showing a moderate linear depolarization ratio of close to

20% and a low lidar ratio of 23 sr is mainly consisting of dry salt particles. A similar signature of enhanced light depolarization in combination with lidar ratios around 25 sr was found by Haarig et al. (2017b) for dried sea salt particles over the remote tropical Atlantic.

The analysis of the backward trajectories for all 45 background aerosol layers (with particle extinction coefficients $<30\,\text{Mm}^{-1}$) revealed that the prevailing, regional airflow was from western to northeastern directions, i.e., from the desiccating saline Lake Urmia in northwestern Iran, the Garabogazköl and Aral Sea to Lake Baikal. There are about 400 desiccating lakes with an area of $<10\,\text{km}^2$ and 50 lakes with an area $>100\,\text{km}^2$ in Central Asia basically fed by glacier melt water. Dry lakes/saline playas are a significant source of atmospheric dust in Asia according to the studies of Abuduwaili et al. (2010) and Issanova et al. (2015). Thus, there is an enormous potential for emission of polluted salt particles. In addition to direct emission, salt dust emitted over past decades, deposited and integrated into the surface soil system is potentially available for re-suspension.

In situ chemical characterization of PM10 (particulate matter with diameters $<10\,\mu$m) based on respective aerosol filter samples taken at the lidar field site at Dushanbe (a few kilometers apart from the city center) during the CADEX field campaign confirm our conclusions. The data showed significantly enhanced concentrations of calcium, potassium, fluoride as well as chloride. The dust calcium/iron ratio was twice as high as that observed for Saharan dust indicating a salt-enriched mineral dust in these particles (Fomba et al., 2019). According to Kandler and Scheuvens (2019) the high carbonate contents internally mixed with silicates observed in the aerosol samples indicate emissions from locations with high evaporation such as dry lake beds. Long-term measurements of the mineralogical composition of dust in Tajikistan also indicate that calcite and potassium mica are among the dominant minerals observed in the aerosols samples (Khodzhakhon et al., 2019). These in situ measurements performed during the CADEX campaigns and beyond confirm that salt-rich dust is present in this region.

Several lidar observations are in agreement with our background aerosol observations. Tesche et al. (2007) found rather low lidar ratios of around 20 sr with lidar in northeastern China during background aerosol conditions (with low extinction coefficients) and an air mass transport from Mongolia and Central Asia in January 2005. Chen et al. (2013) used a multiwavelength Raman lidar (with nitrogen Raman channel at 387 nm) in Kyrgyzstan and combined their observations with spectral sun photometer observations of AOT. In many cases, they retrieved lidar ratios clearly below 30 sr for the visible wavelength range. Dieudonné et al. (2015) also emphasized the comparably low lidar ratios, frequently from 32–40 sr at 355 nm for Russian background aerosol. Khalesifard et al. (2019) and Ghomashi and Khalesifard (2019) found enhanced depolarization ratio up to 0.15 over the desiccating Urmia Lake in northwestern Iran caused by dried salt dust particles.

## 6    Conclusion/Outlook

Within a series of three articles (Hofer et al., 2017, 2020, and this study), we presented our CADEX lidar observations conducted from March 2015 to August 2016 in one of the hot spot regions for expected severe changes in environmental and climate conditions. An advanced continuously running multiwavelength polarization Raman lidar was deployed at Dushanbe, Tajikistan, in Central Asia. In this study, the full spectrum of aerosol mixture from polluted urban haze conditions in winter to pure dust scenarios during heavy dust outbreak periods occurring during the summer half year were discussed. The results

were presented in terms of particle lidar ratios, linear depolarization ratios, and Ångström exponents measured at 2–3 wavelengths in the main aerosol layer typically reaching up to 3–5 km height. Overviews were given separately for each season of the year to cover the seasonal cycle. Surprisingly low lidar ratios for Central Asian dust, frequently below 40 sr and especially during aerosol background conditions were found and may point to a non-negligible contribution of salt dust emissions from
desiccating lakes and the Aralkum desert and re-suspension of deposited salt to the aerosol burden in Central Asia.

The three articles on our CADEX lidar observations can be regarded as the most systematic and densest, vertically resolved data set on Central Asian aerosol conditions published so far. The observations fill a gap in the characterization of aerosols in the center of the northern hemispheric dust belt from western Africa to eastern Asia. For lidar-based aerosol typing, especially with present and future spaceborne lidars such as CALIPSO/CALIOP, EarthCARE/ATLID, and Aeolus/ALADIN, important
data sets in terms of particle linear depolarization ratio, lidar ratio, and Ångström exponent have been established for a region for which no data were available yet. The data are also required to support the basic data analysis of spaceborne lidar missions and harmonization of the overall spaceborne aerosol observations with lidar operated at different laser wavelengths. These data are also of great value for modeling groups dealing with changes of atmospheric, environmental, and future climate forecast.

As an outlook, more studies on environmental conditions in the framework of well-established lidar networks and surface
in situ aerosol measurement stations in the Middle East, Central Asia and eastern Asia would be desirable to improve atmospheric research and environmental forecast (e.g., of severe dust outbreaks). We just covered a period of 1.5 years with aerosol profile observations and need at least further 5-10 years of measurements to obtain a clear (climatological) view on Central Asian aerosol conditions and to derive conclusions on trends in the pollution and dust characteristics. More environment studies would also foster the collaboration with local atmospheric scientists and would upgrade their research potential. It is also
hoped that the use of these observational data by weather bureaus and environmental services would sensitize the population for climate change problems and to better take care of the environment and to avoid further degradation. As a first step, our institute established a new long-term observational field site (an outpost station of EARLINET) at Dushanbe in June 2019, with a new Polly system (now with a diode-pumped laser source). It is planned to run this station over the next 5–10 years to improve aerosol and especially dust monitoring and to support modeling groups by offering continuous, height-resolved
observations in Central Asia. The station will also serve as a ground truth station for satellite missions.

*Financial support.* The CADEX project was funded by the German Federal Ministry of Education and Research (BMBF) in the context of "Partnerships for sustainable problem solving in emerging and developing countries" under the grant number 01DK14014. The construction of a new lidar for permanent observations in Tajikistan is funded by the BMBF under the grant number 01LK1603A. This project
has also received funding from the European Union's Horizon 2020 research and innovation program ACTRIS-2 Integrating Activities (H2020-INFRAIA-2014-2015, grant agreement no. 654109) and from the European FP7 project by the European Union's Seventh Framework Program (FP7/2007-2013) collaborative project BACCHUS (grant agreement no. 603445).

*Competing interests.* The authors declare that they have no conflict of interest.

*Author contributions.* JH and AA wrote the manuscript. JH performed the lidar data analysis supported by HB. KWF performed the chemical aerosol characterization. JH, DA, SFA, and ANM conducted the field experiment with support by RE and UW.

*Special issue statement.* This article is part of the special issue "EARLINET aerosol profiling: contributions to atmospheric and climate research".

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

**Table 1.** Particle optical properties observed with lidar during the CADEX campaign and typical values for the relative uncertainties in the presented layer mean values.

| Parameter | Uncertainty |
|---|---|
| Backscatter coefficient, 355 nm | 5–10% |
| Backscatter coefficient, 532 nm | 5–10% |
| Backscatter coefficient, 1064 nm | 10–20% |
| Extinction coefficient, 355 nm | 10–15% |
| Extinction coefficient, 532 nm | 10–15% |
| Lidar ratio, 355 nm | 15–20% |
| Lidar ratio, 532 nm | 15–20% |
| Linear depolarization ratio, 355 nm | 5% |
| Linear depolarization ratio, 532 nm | 5% |
| Backscatter-related Ångström exponent (355–532 nm) | 10% |
| Backscatter-related Ångström exponent (532–1064 nm) | 20% |
| Extinction-related Ångström exponent (355–532 nm) | 10–20% |

**Table 2.** Mean values $\pm$ SD of pure dust and non-dust lidar ratios ($S_{355}$, $S_{532}$), particle linear depolarization ratios ($\delta_{355}$, $\delta_{532}$) at 355 and 532 nm wavelength, and extinction-related Ångström exponent ($\mathring{a}_{\alpha,355/532}$). N is the number of cases.

| Type | N | $S_{355}$ [sr] | $S_{532}$ [sr] | $\delta_{355}$ | $\delta_{532}$ | $\mathring{a}_{\alpha,355/532}$ |
|---|---|---|---|---|---|---|
| Dust | 17 | 43±3 | 39±4 | 0.24±0.03 | 0.33±0.01 | 0.1±0.2 |
| Non-dust | 7 | 50±10 | 51±10 | 0.02±0.01 | 0.03±0.01 | 1.5±0.3 |

**Table 3.** Seasonal mean(median) values $\pm$ SD of lidar ratios ($S_{355}$, $S_{532}$) and particle linear depolarization ratio ($\delta_{355}$, $\delta_{532}$) at 355 and 532 nm wavelength. The range of the values from minimum to maximum is given in the second row.

| Season | $S_{355}$ [sr] | $S_{532}$ [sr] | $\delta_{355}$ | $\delta_{532}$ |
|---|---|---|---|---|
| MAM | 42(40)±8 | 33(32)±7 | 0.09(0.08)±0.05 | 0.17(0.16)±0.07 |
|  | 28–69 | 21–56 | 0.02–0.26 | 0.03–0.34 |
| JJA | 40(40)±4 | 31(31)±5 | 0.14(0.13)±0.05 | 0.23(0.22)±0.06 |
|  | 28–57 | 19–44 | 0.03–0.29 | 0.08–0.36 |
| SON | 43(43)±5 | 31(30)±5 | 0.10(0.11)±0.04 | 0.19(0.21)±0.07 |
|  | 30–56 | 23–46 | 0.02-0.19 | 0.02–0.29 |
| DJF | 42(36)±13 | 43(39)±14 | 0.05(0.06)±0.02 | 0.09(0.10)±0.05 |
|  | 28–65 | 25–66 | 0.01–0.08 | 0.02–0.19 |

**Table 4.** Seasonal mean(median) values $\pm$ SD of backscatter-related ($\mathring{a}_{\beta,355/532}$, $\mathring{a}_{\beta,532/1064}$) and extinction-related ($\mathring{a}_{\alpha,355/532}$) Ångström exponents . The range of the values from minimum to maximum is given in the second row.

| Season | $\mathring{a}_{\alpha,355/532}$ | $\mathring{a}_{\beta,355/532}$ | $\mathring{a}_{\beta,532/1064}$ |
|---|---|---|---|
| MAM | 1.3(1.4)$\pm$0.6 | 0.7(0.7)$\pm$0.5 | 0.7(0.7)$\pm$0.2 |
| | $-0.6$–2.2 | $-0.4$–1.8 | 0.1–1.4 |
| JJA | 0.9(0.8)$\pm$0.5 | 0.2(0.2)$\pm$0.3 | 0.4(0.4)$\pm$0.2 |
| | $-0.1$–2.4 | $-0.4$–1.2 | 0.1–1.1 |
| SON | 1.2(1.2)$\pm$0.5 | 0.4(0.2)$\pm$0.5 | 0.5(0.4)$\pm$0.2 |
| | 0.3–3.3 | $-0.2$–1.8 | 0.2–1.1 |
| DJF | 1.1(1.1)$\pm$0.4 | 1.1(1.0)$\pm$0.3 | 0.9(0.9)$\pm$0.3 |
| | 0.4–1.8 | 0.6–1.6 | 0.2–1.4 |

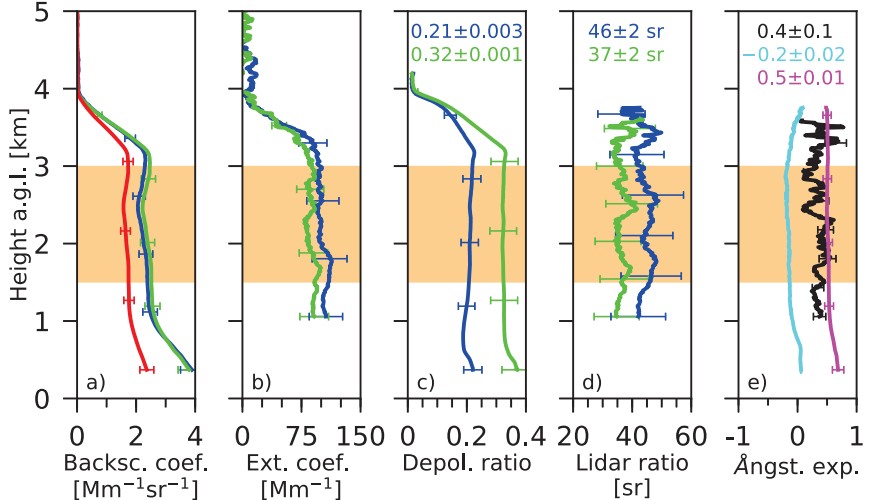

**Figure 1.** Profiles of optical properties measured on 11 June 2015, 18:00–20:59 UTC. (a) Particle backscatter coefficient at 355 nm (blue), and 532 nm (green), and 1064 nm wavelength, (b) particle extinction coefficient at 355 nm (blue) and 532 nm (green), (c) particle linear depolarization ratio at 355 nm (blue) and 532 nm (green) wavelength, (d) lidar ratio at 355 (blue) and 532 nm (green) wavelength, (e) extinction-related Ångström exponent (black), backscatter-related Ångström exponent for the 355–532 nm (light blue) and 532–1064 nm (magenta) wavelength ranges. The horizontal orange area indicates the averaging height range from 1.5–3 km height. All profiles were calculated with the same 743 m vertical smoothing length. Numbers in (c), (d), and (e) show mean values for the orange layer and respective standard deviations. Error bars show the measurement uncertainties in the height profiles.

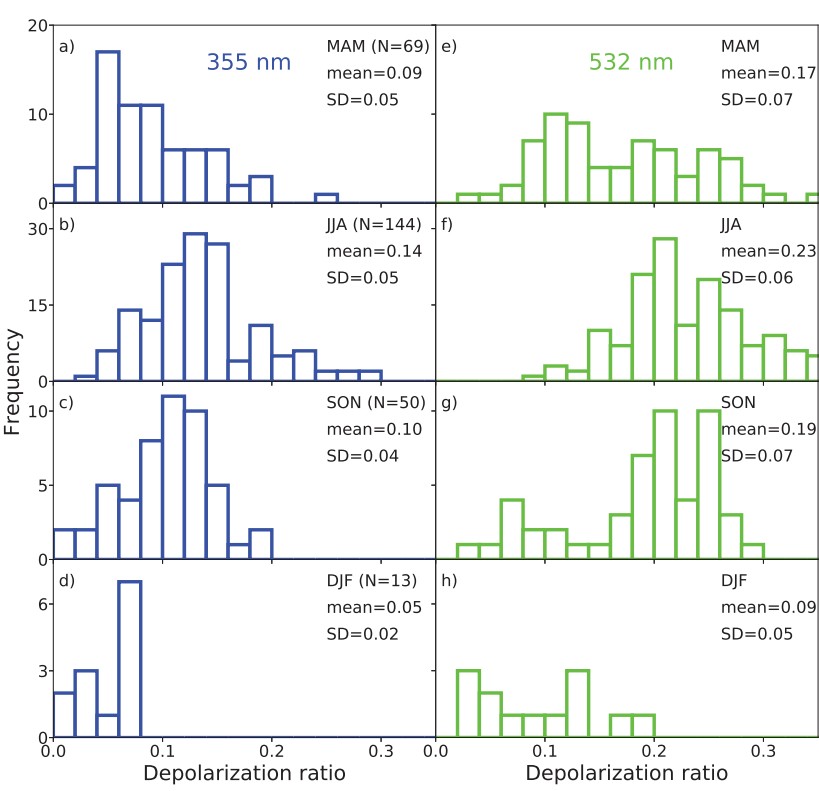

**Figure 2.** Histograms of layer-mean particle linear depolarization ratio at 355 nm (blue, a–d) and 532 nm wavelength (green, d–h) per season for spring (a,e), summer (b,f), autumn (c,g), and winter (d,h). The number of observations per season is given in brackets (a–d). In addition, seasonal mean values and standard deviations are listed.

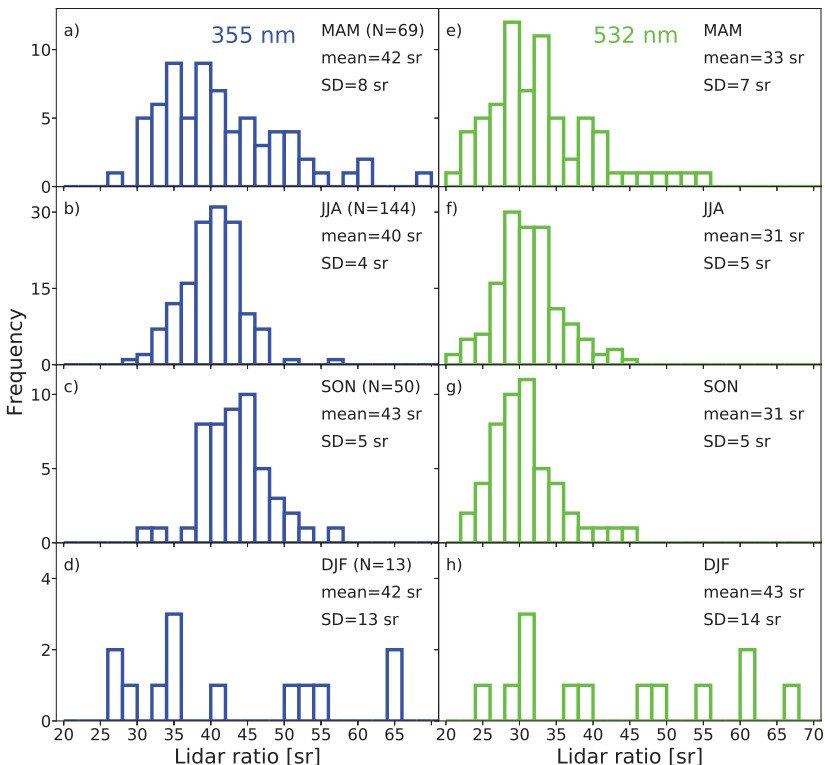

**Figure 3.** Same as Fig. 2, except for the lidar ratios at 355 and 532 nm.

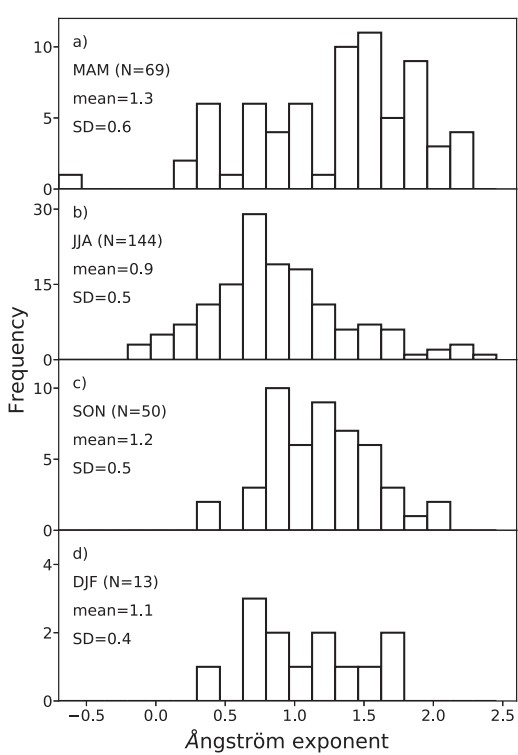

**Figure 4.** Same as Fig. 2, except for the extinction-related Ångström exponent.

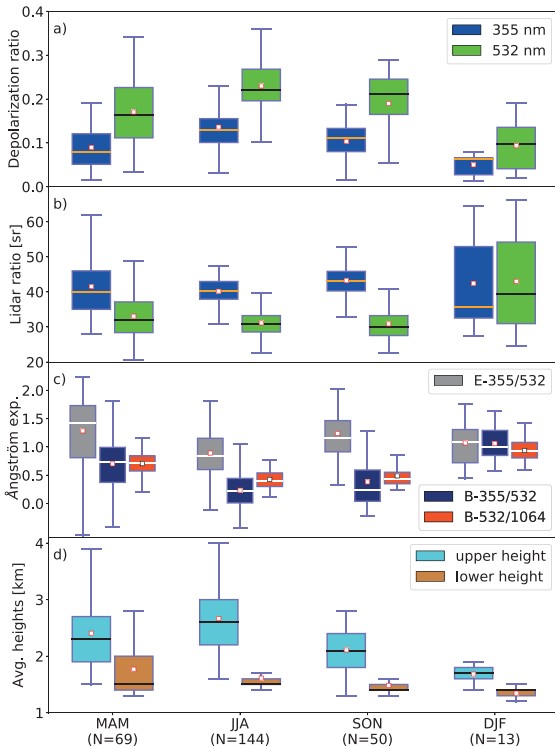

**Figure 5.** Boxplots of layer-mean (a) particle linear depolarization ratio and (b) lidar ratio at 355 and 532 nm wavelength, (c) of extinction-related (E-355/532) and backscatter-related (B-355/532, B-532/1064) Ångström exponents, and (d) of lower and upper boundaries of the respective signal averaging layers for which the optical properties are computed. The statistical results are given for spring (MAM), summer (JJA), autumn (SON), and winter (DJF) with number of observations in brackets. The box indicates the interquartile range IQR), in which 50% of the data accumulate. 75% of the data points are within the range indicated by the whiskers (1.5·IQR).Seasonal mean and median values are given as open squares and horizontal lines, respectively.

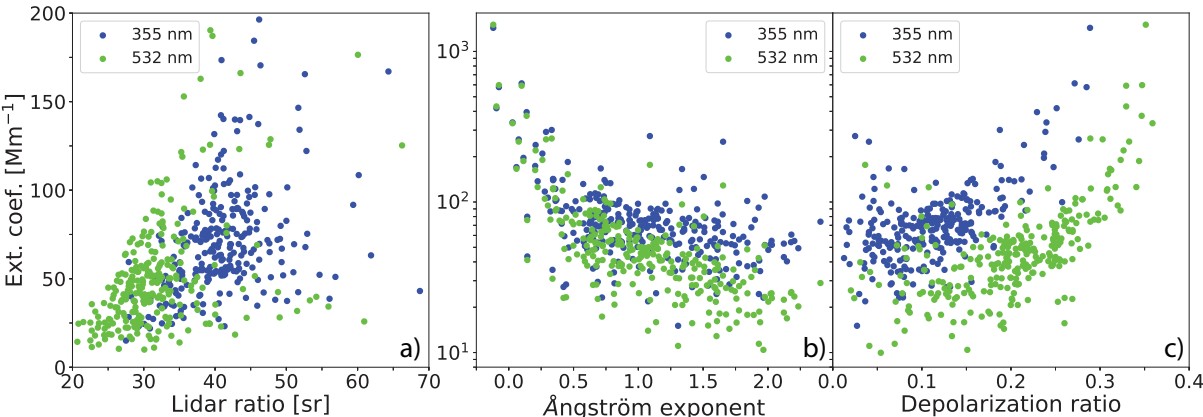

**Figure 6.** Correlation between the particle extinction coefficient at 355 and 532 nm and (a) the 355 and 532 nm particle lidar ratios, (b) the extinction-related Ångström exponent (355–532 nm spectral range, blue and green show the same data set on the x-axis), and (c) 355 and 532 nm particle linear depolarization ratios. All individual nighttime observations of the 1.5-year campaign are considered.

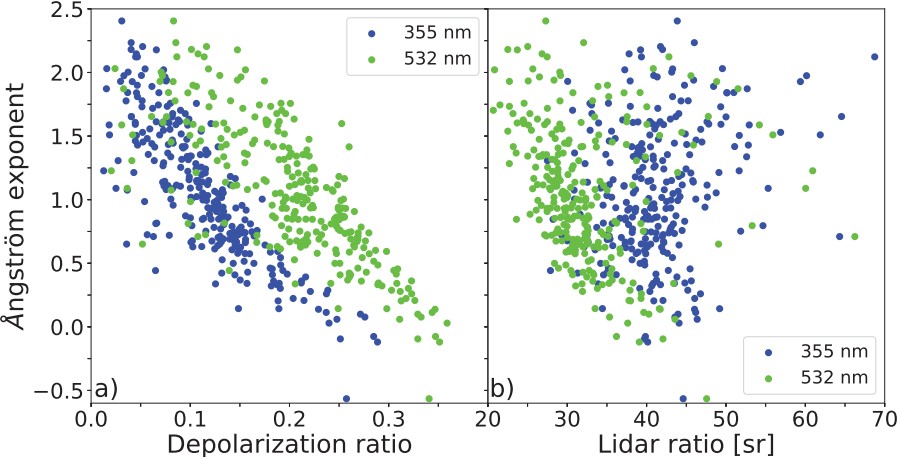

**Figure 7.** Correlation between the extinction-related Ångström exponent (355–532 nm) and (a) the 355 and 532 nm particle linear depolarization ratios and (b) the 355 and 532 nm particle lidar ratios.

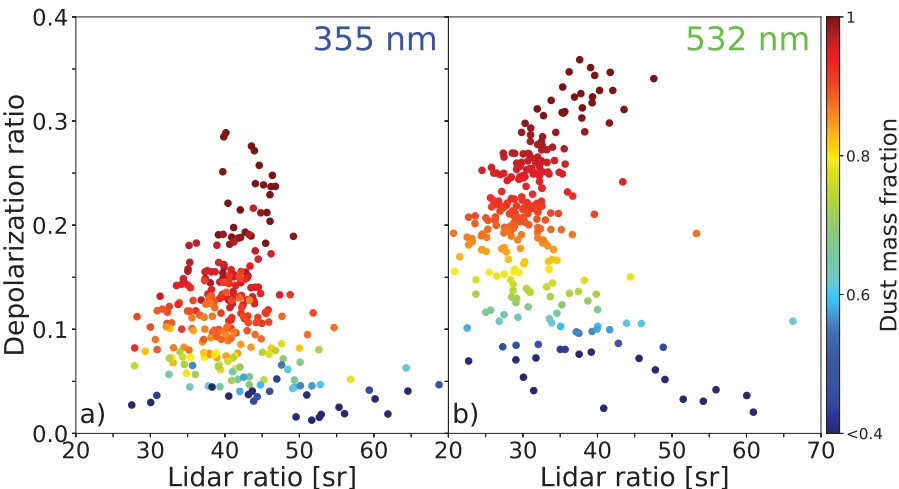

**Figure 8.** Correlation between (a) the 355 nm particle linear depolarization ratio and the 355 nm particle lidar ratio, and (b) the 532 nm particle linear depolarization ratio and the 532 nm particle lidar ratios. The depolarization and lidar ratios are color-coded as a function of dust mass fraction.

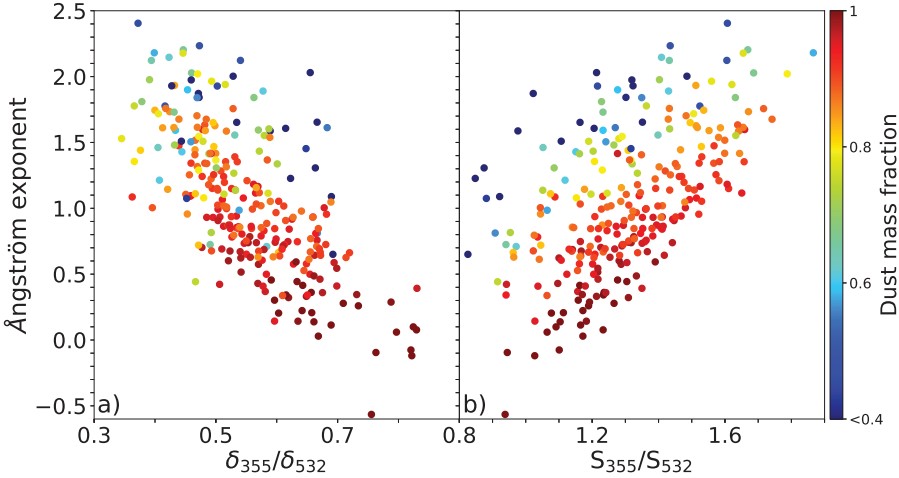

**Figure 9.** Correlation between the extinction-related Ångström exponent and (a) the ratio of depolarization ratios and (b) the ratio of lidar ratios at 355 and 532 nm wavelength. The values are color-coded as a function of dust mass fraction.

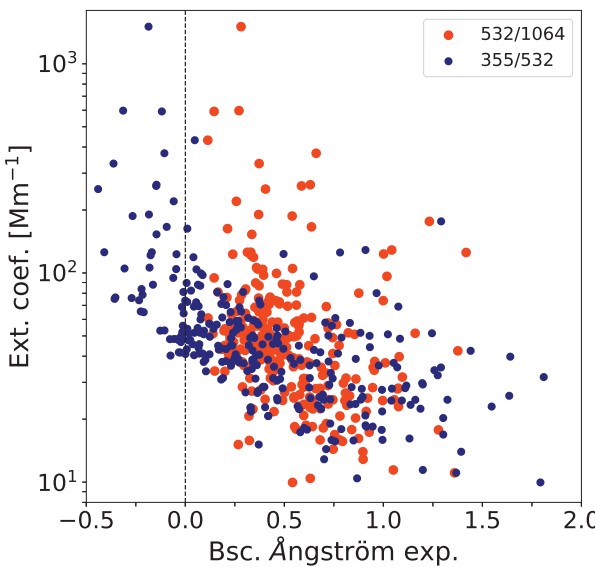

**Figure 10.** Correlation between the backscatter-related Ångström exponents (blue: 355–532 nm spectral range, red: 532–1064 nm spectral range) and the 532 nm particle extinction coefficient.

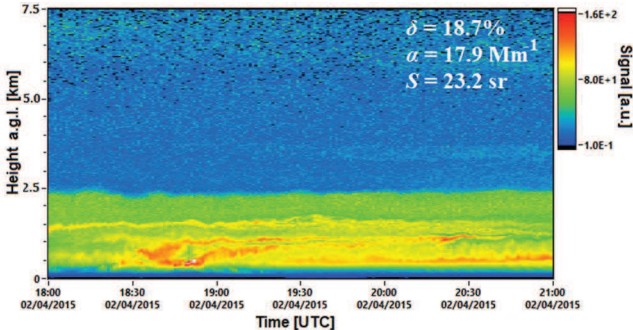

**Figure 11.** 1064 nm range-corrected signal (in arbitrary units) observed with lidar at Dushanbe, Tajikistan, on 2 April 2015, 18:00-21:00 UTC (23:00-02:00 local time). The given numbers for the depolarization ratio $\delta$, the particle extinction coefficient $\alpha$, and the lidar ratio $S$ are mean values for the aerosol layer from 1.6-2.1 km height and the time period from 18:20-19:30 UTC. The incomplete laser-beam receiver-field-of-view overlap prohibits full signal detection up to about 500 m height.

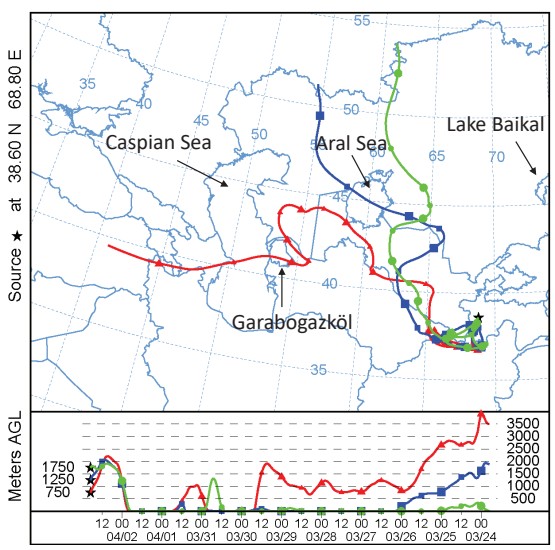

**Figure 12.** 10-day backward trajectories arriving at Dushanbe, Tajikistan, at 750 m (red), 1250 m (blue), and 1750 m height on 2 April 2015, 19:00 UTC. The HYSPLIT model (Hybrid Single Particle Lagrangian Integrated Trajectory Model) (HYSPLIT, 2020; Stein et al., 2015; Rolph, 2016) is applied. Different sources of direct salt dust emissions are indicated. The right half of the Aral Sea is the Aralkum desert.