# Peer review of "Optical properties of Central Asian aerosol relevant for spaceborne lidar applications and aerosol typing at 355 and 532 nm"

_Atmospheric Chemistry and Physics, 2020_

## Referee Comment (RC1) · Anonymous Referee #2 · 23 Apr 2020

The manuscript titled "Optical properties of Central Asian aerosol relevant for space-borne lidar applications and aerosol typing at 355 and 532 nm" by Hofer et al., present a dataset of aerosol properties derived by an advanced lidar system, that the authors claim were never available in this geographical location (Dushanbe, Tajikistan, Central Asia). The dataset show a large range of optical properties reflecting the complex aerosol mixture of background aerosol (mainly soil dust and salt dust from over 400 desiccating lakes in Central Asia and the Aralkum desert), long-range transport of mineral dust from the deserts in Middle East and from the Sahara, regional desert dust, local and regional anthropogenic haze and fire smoke. The authors focus the study on those properties that are used in aerosol-typing efforts with present and future spaceborne lidars. The manuscript is well-written and it can be published in ACP. Although there are several issues and technical comments that can improve it.

Firstly, the manuscript is the third about the campaign, as the authors mention in the conclusions section, and some analysis is based in the two previous articles. Since this isn't clearly established by including in the title "part I, II and III", to inform the reader about previous reads, the manuscript should be self-contained. For instance, the authors explain the variability of the aerosol properties by complex aerosol mixtures (pag. 5, lines 12-13). Further details must be included about the procedure to do so, despite the full explanation may be provided in other article. In the same line, the most relevant results regarding aerosol typing are based on the "dust fraction", that the authors don't describe, just mention (pag. 8, line 30-31) the reference where its determination is provided. A brief description should be included with the same aim to make the manuscript self-contained.

Secondly, the oddly low lidar ratios found during background conditions are attributed to salt dust emissions from desiccating lakes (pag. 10, lines 18-20) without any evidence. Such conclusion is not properly supported by experimental data and a better data analysis, identifying temporal situations when the low lidar ratios are observed, supported with additional information, as in-situ measurements or backtrajectory analysis, must be studied. As it stands now, it seems just a speculation.

Finally, some technical issues: The polly system detects cross-polarized and total backscatter signals (pag. 3, lines 21-25) instead of cross- and co-polarized components. Why is the total backscatter signal instead of the co-polarized signal component detected?. Although the details might be presented in the mentioned references, a brief explanation would be useful.

The number of aerosol profiles available (pag. 4, line 23) isn't clear. There are 487 days with data and 276 of those with data at night. But this 328 are nighttime data, some from the same night?, or daytime profiles are also included, in that case, why

they don't add up to 487?

In page 2, line 26, a database is mentioned but it is not clear if it is a project of the authors or they refer to a general database being collected worldwide by the scientific community. Please be more specific, including relevant references.

Figure 8, explained in pag. 8, lines 29-35, shows another relevant feature, the decrease in spread in the 532 nm lidar ratio when depolarization values increase from <0.1 to 0.2. The figure shows values between 20-30 sr at about 0.2, that later increase to 30-40 sr as depolarization increases to 0.3. It doesn't occur to the LR@355nm. What explanation can the authors provide to that feature?

Figure 1 X-axis labels are hard to read, it would be clearer to separate each graphs by a space that allows the last x-axis label of each graph to be shown.

---

## Referee Comment (RC2) · Ali Omar (Referee) · 11 May 2020

The paper presents optical properties of dust in the Central Asian region. This region has been relatively understudied – this is the first comprehensive data set for this region in a series of papers by the same group of authors. While the authors should be commended for providing this data set, I think it is important to note that the data set is for only one year and may not be universally representative for this region. The authors have engaged local scientists (co-authors on the paper) that will help to build the capacity to do this work, so I hope that we will get a longer data set over many years. The dataset is important for identifying aerosol types using objective criteria

and will be a useful resource for future space-based lidar missions.

There is a ground station in Dushanbe that has been making some measurements in the past. Does this station have any in-situ measurements that includes filter-based measurements? If there are filter based measurements at Dushanbe, these may reveal the composition of the particles in the region and can definitively identify salt particles as the particles with high depolarization ratios and low extinction to backscatter ratios. Otherwise, there is no direct evidence of salt particles beyond the circumstantial evidence in Section 4 of the paper.

In addition, trajectory studies may help identify the source of the particles and trace them back to the desiccating salt lakes that the authors have identified as possible sources of the salt particles.

The Polly technique measures extinction unambiguously at 355 nm and 532 nm. Does it also retrieve extinction at 1064 nm? If it does not, why is the paper referring to 1064 nm backscatter. What method is used to retrieve particulate 1064 nm backscatter. If the method is in another paper, it may be useful to include it in this paper.

---

## Author Comment (AC1) · 17 Jun 2020

Our reply letter is attached.

Please also note the supplement to this comment:
https://www.atmos-chem-phys-discuss.net/acp-2020-258/acp-2020-258-AC1-supplement.pdf
* * *

---

## Author Comment (AC2) · 17 Jun 2020

Dear Editor, Dear Reviewers!

Thank you for taking the time to review this paper and to provide use with carefully elaborated valuable advices. We considered all of them. They clearly improved the paper.

One of the most important improvements is that we created a new Section 5 (on the potential salt dust impact on background aerosol conditions). Here, we discuss results of a chemical aerosol characterization and HYSPLIT trajectory analysis and show a measurement case (new Fig.11) together with HYSPLIT backward trajectories (Fig.12) supporting our hypothesis that salt dust has a strong impact on background aerosol conditions.

**Our answers (point-by-point reply) are given in RED and BOLD.** Significant changes in the manuscript are given in red as well.

**Reviewer #1 (Ali Omar)**

The paper presents optical properties of dust in the Central Asian region. This region has been relatively understudied – this is the first comprehensive data set for this regionin a series of papers by the same group of authors. While the authors should be commended for providing this data set, I think it is important to note that the data set is for only one year and may not be universally representative for this region. The authors have engaged local scientists (co-authors on the paper) that will help to build the capacity to do this work, so I hope that we will get a longer data set over many years. The dataset is important for identifying aerosol types using objective criteria and will be a useful resource for future space-based lidar missions.

**We agree with the statement to emphasize that the data set is for only one year and may not be universally representative for this region.**

**We state that in the outlook part to conclude that more observations over a longer period is required.**

There is a ground station in Dushanbe that has been making some measurements in the past. Does this station have any in-situ measurements that includes filter-based measurements? If there are filter based measurements at Dushanbe, these may reveal the composition of the particles in the region and can definitively identify salt particles as the particles with high depolarization ratios and low extinction to backscatter ratios. Otherwise, there is no direct evidence of salt particles beyond the circumstantial evidence in Section 4 of the paper.

In addition, trajectory studies may help identify the source of the particles and trace them back to the desiccating salt lakes that the authors have identified as possiblesources of the salt particles.

**No, the local scientists did not have the option to perform in situ observations. That was the reason that TROPOS installed instrumentation for a chemical characterization of aerosols. So, we had our own filter probe sampling unit at the lidar site at Dushanbe and we analysed the data now regarding salt-impact signatures. Wadinga Fomba from TROPOS performed this analysis and the conclusions are given in Section 5 (new section on the potential salt dust impact). He detected a clear signature of salt-enriched aerosol and supports our hypothesis that salt dust is a background aerosol component. Wadinga Fomba is added as co-author.**

**We also performed HYSPLIT backward trajectories for all 45 cases (layers) with background aerosol signatures, i.e., with low extinction (background), moderate depolarization and low lidar ratio. And most of them showed slow advection (low**

**impact on long range transport) and all of them came from westerly to northeasterly directions, and thus from areas with a large number of small to large desiccating lakes. All this is stated in a new section (Section 5).**

**For one specific case (2 April 2015) we show the layer (height-time display in terms of range-corrected 1064 nm, Fig. 11) and the corresponding backward trajectories (Fig.12) arriving from the Aralkum desert and the saline Garabogazköl.**

The Polly technique measures extinction unambiguously at 355 nm and 532 nm. Does it also retrieve extinction at 1064 nm? If it does not, why is the paper referring to 1064nm backscatter. What method is used to retrieve particulate 1064 nm backscatter. Ifthe method is in another paper, it may be useful to include it in this paper.

**We analyze all signal profiles we measure. The laser emits 1064, 532, and 355 nm, so we measure the respective backscatter signals. We do not have a Raman channel close to 1064 nm, therefore we have no option to compute 1064 nm extinction coefficients. However, 1064 nm is at least useful to determine the 532-1064nm backscatter Angstroem exponent to see the impact of coarse particles on the optical properties.**

**As explained on page 3 now (Sect. 2), we use the 1064-to-607-nm signal ratioto obtain the backscatter coefficient at 1064nm. This is the classical Raman backscatter retrieval approach. We take care of the different aerosol extinction effects at 532, 607, and 1064 nm. We provide Mattis et al. (2004) as reference. In this paper, this method was mentioned and applied for the first time.**

**Reviewer #2:**

The manuscript titled "Optical properties of Central Asian aerosol relevant for space-borne lidar applications and aerosol typing at 355 and 532 nm" by Hofer et al., present a dataset of aerosol properties derived by an advanced lidar system, that the authors claim were never available in this geographical location (Dushanbe, Tajikistan, Central Asia). The dataset show a large range of optical properties reflecting the complex aerosol mixture of background aerosol (mainly soil dust and salt dust from over 400 desiccating lakes in Central Asia and the Aralkum desert), long-range transport of min-eral dust from the deserts in Middle East and from the Sahara, regional desert dust,local and regional anthropogenic haze and fire smoke. The authors focus the study on those properties that are used in aerosol-typing efforts with present and future spaceborne lidars. The manuscript is well-written and it can be published in ACP. Although there are several issues and technical comments that can improve it.

**Ok!**

Firstly, the manuscript is the third about the campaign, as the authors mention in the conclusions section, and some analysis is based in the two previous articles. Since this isn't clearly established by including in the title "part I, II and III", to inform the reader about previous reads, the manuscript should be self-contained. For instance, the authors explain the variability of the aerosol properties by complex aerosol mixtures (page. 5, lines 12-13). Further details must be included about the procedure to do so, despite the full explanation may be provided in other article.

**We agree, and we provide now a new paragraph on aerosol sources in Central Asia and impacts as given in the paper of Rupakheti et al. (2019). These authors nicely describe the aerosol conditions in this region of the world (see page 5, Section 3.2).**

In the same line, the most relevant results regarding aerosol typing are based on the "dust fraction", that the authors don't describe, just mention (page. 8, line 30-31) the reference where its determination is provided. A brief description should be included with the same aim to make the manuscript self-contained.

**We provide the definition of the dust mass fraction (ratio of dust-to-total particle mass concentration) now in Sect. 2 (page 3) and also explain how we obtained it from the lidar measurements (page 4, last paragraph at the end of Sect. 2).**

Secondly, the oddly low lidar ratios found during background conditions are attributed to salt dust emissions from desiccating lakes (pag. 10, lines 18-20) without any evidence. Such conclusion is not properly supported by experimental data and a better data analysis, identifying temporal situations when the low lidar ratios are observed, supported with additional information, as in-situ measurements or back trajectory analysis, must be studied. As it stands now, it seems just a speculation.

**We agree and enlarged the data analysis and the discussion accordingly. We introduced a new section (Sect. 5) to better highlight this background-aerosol discussion. And added two more figures (Fig. 11 and Fig. 12).**

**Regarding in situ observations: TROPOS installed instrumentation for a chemical characterization of aerosols during the CADEX field campaign from March 2015 to August 2016. We had our own filter probe sampling unit at the lidar site at Dushanbe and we analysed the data regarding salt-impact signatures. Wadinga Fomba from TROPOS performed this analysis and the conclusions are given in the new Section 5. He detected a clear signature of salt-enriched aerosol and supports our hypothesis that salt dust is a background aerosol component. Wadinga Fomba is now added as co-author.**

**Regarding backward trajectories: We performed HYSPLIT backward trajectories for all 45 cases (layers) with background aerosol signatures, i.e., with low extinction (background), moderate depolarization and low lidar ratio. And most of the trajectories showed regional advection (low impact of long range transport) and all of them came from westerly to northeasterly directions and thus from areas with a large number of small to large desiccating lakes. All this is stated now in the new section (Section 5).**

**For one specific case (2 April 2015) we show the layer (height-time display in terms of range-corrected 1064 nm, Fig. 11) and the corresponding backward trajectories (Fig.12) arriving from the Aralkum desert and the saline Garabogazköl.**

Finally, some technical issues: The Polly system detects cross-polarized and total backscatter signals (pag. 3, lines 21-25) instead of cross- and co-polarized components. Why is the total backscatter signal instead of the co-polarized signal component detected?. Although the details might be presented in the mentioned references, a brief explanation would be useful.

**We agree and this statement motivated us to provide more explanation in Section 2 (page 4): The measurement of cross and total backscatter components is a compromise. By measuring the cross and co polarized signal components the optimum solution for depolarization ratios is obtained. But by measuring the total backscatter component, the optimum solution for the lidar ratio is obtained. And for us, the lidar ratio retrieval has the highest priority (since 20 years…) and therefore we measure the total backscatter signal rather than the cross-and co-polarized backscatter signal components from which the total backscatter signal must then be**

**constructed. By measuring the total lidar return signal we keep the uncertainties in the lidar ratio calculations as small as possible (see Sect. 2, page 4).**

The number of aerosol profiles available (pag. 4, line 23) isn't clear. There are 487days with data and 276 of those with data at night. But this 328 are nighttime data,some from the same night?, or daytime profiles are also included, in that case, why they don't add up to 487?

**We clarified these confusing statements. We have all in all 487 night time profiles. Then we have 276 cases with the full set of Raman lidar profiles (extinction, backscatter, lidar ratio). But we had the chance … in further 52 nights to determine at least the particle backscatter profile (by using the Raman method) …. and these 276+52=328 profiles are the basis for the layer structure analysis (see Sect. 3, page 5).**

In page 2, line 26, a database is mentioned but it is not clear if it is a project of the authors or they refer to a general database being collected worldwide by the scientific community. Please be more specific, including relevant references.

**We now write: … to the steadily growing worldwide aerosol-typing data base**

Figure 8, explained in pag. 8, lines 29-35, shows another relevant feature, the decrease in spread in the 532 nm lidar ratio when depolarization values increase from <0.1 to0.2. The figure shows values between 20-30 sr at about 0.2, that later increase to30-40 sr as depolarization increases to 0.3. It doesn't occur to the LR@355nm. What explanation can the authors provide to that feature?

**We explain that now on pages 9-10 in Sect.4: The curved feature in Fig. 8b with higher lidar ratios for small and large 532 nm depolarization ratios is in principle also found for 355 nm, however the range of observed depolarization ratio is smaller so that the curved feature is compressed and a clear minimum of the 355 nm lidar ratio for moderate depolarization ratios around 0.1 is not visible. Furthermore, the optical properties at 355 nm are dominated by scattering and absorption by fine-mode aerosol (especially by anthropogenic haze) and are less influenced by scattering by coarse-mode (desert or salt) dust particles than the ones for the 532 nm wavelength. Thus, the background aerosol effect shows up more pronounced for 532 nm.**

Figure 1 X-axis labels are hard to read, it would be clearer to separate each graphs bya space that allows the last x-axis label of each graph to be shown.

**The figure is changed accordingly.**